# App-Based Lifestyle Intervention (PINK! Coach) in Breast Cancer Patients—A Real-World-Data Analysis

**DOI:** 10.3390/cancers16051020

**Published:** 2024-02-29

**Authors:** Josefine Wolff, Martin Smollich, Pia Wuelfing, Jack Mitchell, Rachel Wuerstlein, Nadia Harbeck, Freerk Baumann

**Affiliations:** 1Breast Center, Department of Gynecology and Obstetrics, Comprehensive Cancer Center, LMU University Hospital, 80336 Munich, Germanynadia.harbeck@med.uni-muenchen.de (N.H.); 2Institute of Nutritional Medicine, University Hospital Schleswig-Holstein, 23538 Luebeck, Germany; martin.smollich@uksh.de; 3Department Clinical Research, PINK! gegen Brustkrebs GmbH, 20251 Hamburg, Germanyjack.mitchell@pink-brustkrebs.de (J.M.); 4Department I of Internal Medicine, University of Cologne, 50923 Cologne, Germany

**Keywords:** breast cancer, mhealth app, ehealth, psychological distress, coaching

## Abstract

**Simple Summary:**

The presented data provides intriguing insights into the users of the PINK! Coach app and the impact of this usage in regards to body mass index (BMI) and physical activity. At the current time, there are only a few effective concepts for encouraging all breast cancer patients to engage in moderate physical activity and reduce body weight. Often, these concepts apply to selected patient groups. The data presented here include all age groups, tumor stages, and therapies, providing an initial insight into a comprehensive approach. Data over an even longer period would be one way to better contextualize the results in current research.

**Abstract:**

Introduction: Overweight and a lack of physical activity not only increase the risk of recurrence in breast cancer patients but also negatively impact overall and long-term survival, as well as quality of life. The results presented here are the first real-world data from the DiGA PINK! Coach examining the physical activity and BMI of app users. Based on the literature, an approximate weight gain of 10% over 6 months and a decrease in physical activity can be expected. The purpose of this study is to retrospectively investigate the effects of the PINK! Coach in a real-world setting on patients’ BMI and physical activity level during acute therapies. such as chemotherapy (CHT) and antihormone therapy (AHT). Material and Methods: The PINK! Coach app accompanies breast cancer patients during and after acute therapy to bring about a sustainable lifestyle change. The patients are encouraged to establish a healthy diet, become physically active, and make informed decisions. In this study, real-world data from the app were analyzed over 6 months from baseline to T1 (after 12 weeks) and T2 (after 24 weeks). The patients were under acute therapy or in follow-up care receiving either CHT or AHT. Results: The analyzed data indicate that all patients were able to maintain a consistent BMI over 6 months independent of pre-defined subgroups such as AHT, CHT, or BMI subgroups. In the subgroup of patients undergoing AHT, overweight patients were even able to significantly reduce their BMI by 1-score-point over 6 months (*p* < 0.01). The subgroup of patients undergoing CHT also showed an significant overall reduction in BMI (*p* = 0.01). All patients were also able to significantly increase their daily step count as well as their physical activity minutes per day. After the first 12 weeks, 41.4% of patients experienced weight gain, 33.4% were able to maintain their weight, and 24.2% reduced their weight. Conclusion: The presented data provides intriguing insights into the users of the PINK! Coach app and the impact of this usage in regards to BMI and physical activity. At the current time, there are only a few effective concepts for encouraging all breast cancer patients to engage in moderate physical activity and reduce body weight. Often, these concepts apply to selected patient groups. The data presented here include all age groups, tumor stages, and therapies, providing an initial insight into a comprehensive approach. Data over an even longer period would be one way to better contextualize the results in current research.

## 1. Introduction

Breast cancer is the most prevalent malignant neoplasm among women, affecting one in every eight women [1]. It is also the second most common cause of cancer-related deaths in women [1,2,3]. Breast cancer is linked to various risk factors, which are commonly categorized into non-modifiable factors, such as BRCA1/2, and modifiable factors, such as a lack of physical activity and obesity. Approximately 25% of cancer cases globally can be attributed to excess weight and a sedentary lifestyle [4].

Furthermore, physical activity is linked to a reduced risk of breast cancer recurrence [4,5,6,7,8,9] and improved survival [3,4,5,6,7]. It is probable that these diminished risks result from multiple interconnected biological pathways, including sex hormones, inflammation, insulin resistance and levels, oxidative stress, and adipokines [10].

Both acute and chronic physical activity exert a wide range of physiological effects on immune cells, impacting both their quantity and functionality. Generally, moderate physical activity has been linked to heightened T cell proliferation [5,11], reduced levels of senescent T cells [6], decreased inflammatory cytokines [5,6,11], and increased NK cell cytotoxicity [12]. The augmentation in circulating cell numbers and cytotoxicity could play a crucial role in immune cell surveillance against cancer. While individual instances of physical activity can directly enhance the quantity and function of cytotoxic cells, it is the cumulative impact of regular physical activity that has been correlated with a decreased risk of cancer in humans [4,5].

The guidelines for the treatment and follow-up of breast cancer patients recommend moderate physical activity as part of the post-treatment care. This corresponds to approximately 300 min of activity per week. This goal is also integrated into the PINK! Coach App [13,14,15,16,17].

Studies found that multiple factors determine physical activity compliance rates in breast cancer survivors. Interventions that encourage action and coping planning, while reducing barriers in the context of addressing functional limitations, may increase physical activity compliance rates. Educating patients about the advantages of consistent physical activity holds significant importance for all cancer survivors. This becomes particularly crucial to emphasize post-treatment completion, fostering healthy habits during this transitional phase. Survivors with lower educational backgrounds and comorbid conditions may exhibit lower adherence to physical activity guidelines [18,19,20]. 

Equally recognized is the correlation between physical activity, nutrition, and BMI/obesity [21,22,23]. Obesity represents a global health epidemic, affecting over 500 million adults worldwide [24]. Among post-menopausal women, obesity is a well-established risk factor for the development of breast cancer, and it has been associated with an increased risk of breast cancer recurrence and poorer survival rates in both pre- and post-menopausal patients [7,21,22,23]. Initial results from randomized, controlled trials indicate that lifestyle modifications can positively impact biomarkers related to breast cancer progression and overall survival [21,22,23]. The BMI is a central anthropometric measure used to classify an individual’s body weight in relation to their height. It serves as a key parameter for the nutritional classification of underweight (BMI < 18 kg/m^2^), normal weight (BMI 18–25 kg/m^2^), overweight (BMI 25–30 kg/m^2^), and obesity (BMI > 30 kg/m^2^). Overweight and obesity are established risk factors for non-communicable diseases, and the risk of developing post-menopausal breast cancer significantly increases with an increased BMI [21,22,23]. 

In the presence of obesity (BMI ≥ 30 kg/m^2^), not only does the risk of recurrence increase, but there is also an elevated risk of comorbidities such as cardiovascular diseases, lipid metabolism disorders, and type 2 diabetes. Current studies also indicate that an elevated BMI in patients with BRCA1/2 mutations is associated with increased aromatase expression in breast tissue. Aromatase catalyzes the synthesis of estrogen, which is in turn associated with the growth of breast cancer cells. Additionally, overweight and obesity are considered risk factors for hormone receptor-positive breast cancer. A high BMI leads to subclinical inflammation in visceral and subcutaneous adipose tissue, characterized by necrotic adipocytes surrounded by macrophages forming specific structures known as crown-like structures. The so-called obesity-inflammation axis is under investigation in current clinical studies and plays a relevant role in the initiation, progression, and clinical prognosis of breast cancer [25,26]. Studies also indicate that approximately one-third of the patients are of normal weight, one-third are overweight, and one-third are obese [27].

This is why in the recurrence prevention of breast cancer, the avoidance or reduction of overweight and obesity play a crucial role [25]. Additionally, it is known from studies that patients undergoing endocrine therapy experience weight gain [23,27,28]. This is associated with a change in lipid metabolism due to hormonal treatment [24,25,26,29]. Weight gain is also expected during CHT, as patients receive cortisone and may engage in less physical activity due to therapy-related side effects [29,30,31].

Based on these research findings, it is essential for cancer patients undergoing active therapy and survivors to make sustainable changes to their lifestyle. The importance of making well-informed lifestyle choices becomes particularly evident for cancer survivors as they anticipate the successful completion of therapy and explore self-care strategies to enhance their long-term outcomes [25,26]. For numerous individuals who have survived cancer over the long term, prioritizing healthy weight management, adopting a nutritious diet, and embracing a physically active lifestyle are essential measures aimed at preventing long-term side effects such as fatigue, recurrence, second primary cancers, and other chronic diseases. Breast cancer patients tend to engage in insufficient physical activity during and after therapy. This is often attributed to the side effects experienced during the treatment. In the post-treatment phase, additional side effects may arise, and patients receive less frequent monitoring. Currently, there are few effective concepts aimed at achieving a sustained increase in physical activity [14,29,30,31,32,33,34,35,36,37,38]. To understand the mechanism behind long-term lifestyle change, various models can be considered. There are different models including the Health Belief Model, Social Cognitive Theory, and the Theory of Planned Behavior, each emphasizing different factors influencing behavior change, such as perceived susceptibility, self-efficacy, and social norms.

The “Circle of Change” or “Transtheoretical Model” is widely used in health psychology and behavior change interventions, including those aimed at lifestyle habits like physical activity, healthy eating, and smoking cessation. This model includes six steps that happen when patients change their lifestyle habits: Precontemplation, Contemplation, Preparation, Action, Maintenance, and Termination [39]. New strategies are necessary to support all patients in a low-threshold manner. To make sustainable lifestyle changes, most patients require support, guidance, and motivation. 

However, hospital care does not typically include such coaching. To fill this healthcare void and enhance assistance for breast cancer patients and survivors throughout various stages of treatment, digital app-based solutions offer a promising and cost-effective avenue [40,41,42,43]. 

The digital health application PINK! Coach is an app-based coaching tool designed for breast cancer patients in therapy and survivors [44]. The app is approved as the first Digital Health Application (DiGA) for breast cancer patients and can be prescribed by any doctor in Germany. 

In this study, the initial real-world data from the PINK! Coach app are being analyzed to retrospectively investigate the effects of PINK! Coach in a real-world setting on patients’ BMI and physical activity level during acute therapy (CHT, AHT).

## 2. Materials and Methods

This study is a retrospective observational study. Patients who use the PINK! Coach App consent to the anonymized analysis of their data when downloading the app.

### 2.1. Participants and Procedure

The following Figure 1 illustrates data extraction based on inclusion and exclusion criteria. All patients who were prescribed the app as a DiGA and used it for a minimum of 3 months within the period from July 2022 to November 2023 were included. 

At the time of export, 1752 records were identified in the app’s database. Among these, 215 were excluded due to non-usage of the app. Additionally, 761 did not meet the criterion of having both a baseline and T1 follow-up data point. Records of patients who did not actively use the app (non-users) were not included. Active usage was defined as follows: a patient is considered an active user if, at the time of export, they have used the app at least 3 times in the last 14 days. Usage, in this case, involves logging in with at least one additional action in the app.

### 2.2. Intervention

The PINK! Coach App provides personalized, evidence-based therapy and the management of side effects. It includes mindfulness-based stress reduction, educational resources on nutrition and psychology, tracking of physical activity, and motivational exercises to facilitate the integration of sustainable lifestyle changes into the daily routines of breast cancer patients. PINK! Coach is designed to offer support to individuals with breast cancer at every stage of the disease and treatment, empowering them to actively participate in their therapy and post-treatment care. This empowerment is irrespective of factors such as the stage of the disease, type of therapy, age, place of residence, or the specific clinic where the patient is receiving or has received treatment. Utilizing PINK! Coach provides patients with personalized coaching that is flexible in terms of time and location.

To motivate patients towards more physical activity and a healthier lifestyle, the PINK! Coach App suggests personalized, daily goals for them to achieve. These goals include behavioral changes such as avoiding specific unhealthy foods, short meditations for stress reduction, or a step goal. Additionally, it documents their progress and sends push notifications to encourage long-term engagement. 

Each patient received the app by prescription. Patients used the app at their own discretion and with varying intensity during their acute therapy or during the follow-up phase. There were no further guidelines for usage. The functionalities of the app are explained during an onboarding process after the download, and patients are guided step by step through the app.

### 2.3. Outcome Measures 

To evaluate the real-world effect of PINK! Coach on patients with breast cancer in acute therapy and survivorship, a retrospective observational study was conducted. The data were retrospectively analyzed to gain initial insights into the effects of the app. All analyses were conducted exploratively.

The aim was to examine the trajectory of physical activity and the BMI of the patients over 6 months with data collection at baseline (T0), after 12 weeks (T1), and after 24 weeks (T2). Based on the literature, it is assumed that BMI increases over time. Particularly in the subgroups AHT and CHT, an increase in BMI is expected. Furthermore, therapy subgroups and BMI categories were examined. Physical activity was measured using various parameters: average steps per day and activity minutes per day were determined. The weight and height for calculating the BMI and the activity minutes were self-reported by the patients. 

### 2.4. Definition of Data 

All data were analyzed in anonymized form. Only datasets from patients who consented to the anonymized analysis of their data at the time of download were included. According to the institution’s guidelines, consultation with the ethics committee is not required for purely retrospective studies.

At the time of downloading the app, patients provide information about the tumor biology, the tumor stage, therapy status (undergoing therapy/in follow-up), the therapy they have received or will receive (not recommended, planned, under acute therapy, therapy completed, unknown), as well as their year of birth and weight. Data on the type of therapy and weight are re-evaluated in 4-week intervals. The number of steps is determined by the integrated step counter in the app. Patients are encouraged to take their phones with them during physical activity. The app calculates activity minutes based on the information provided by patients regarding the type and duration of sports activities performed on a given day.

### 2.5. Statistical Analysis

The modified full analysis set (mFAS) contained all adherent patients with data points at baseline and after 12 weeks. The significance level was set at *p* < 0.05. Data normality was evaluated with a chi-square test. The primary analysis was the difference in changes of mean BMI, steps per day, and physical activity minutes per week from baseline to T2 (24 weeks) overall and for the predefined CHT and AHT subgroups, as well as the BMI subgroups. To examine whether the change in mean BMI from T0 to T2 (24 weeks) is dependent on the covariates (confounding factors) of the tumor stage, early breast cancer (EBC) and metastatic breast cancer (MBC), CHT-status, AHT-status, and therapy status (in acute therapy/follow-up), a repeated measures ANCOVA was conducted. Since the test for sphericity yielded a significant result, the Greenhouse–Geisser correction was applied. Subsequently, the significant factors were re-examined in a post-hoc test. 

For sensitivity analysis, different imputation algorithms were used to deal with missing values at post-assessment. As secondary analyses, the changes in mean steps per day and mean physical activity minutes were examined overall and within the subgroups of patients under acute CHT and AHT.

The demographic, medical history, and outcome variables were described using frequency and descriptive statistics [mean  ±  standard deviation (SD)]. Analyses were performed using SPSS Version 29.0.1.0.

## 3. Results 

### 3.1. Descriptive Statistics

Initially, we examined the group of patients who were not included in the analyses (*n* = 976) versus the mFAS (*n* = 776). No significant differences were observed between these two sets in terms of clinical parameters such as age, disease stage, tumor stage, therapy status, or CHT- and AHT-status. 

The following table (Table 1) illustrates the characteristics of the examined mFAS with regards to age distribution, tumor stage, therapy status, and the receipt of AHT or CHT. 

The distribution of these factors reflects clinical reality. Approximately half of all patients receive CHT [3], and around 70% undergo endocrine therapy [45,46]. In the examined cohort, approximately 42% received CHT, and around 73% received AHT. The group of patients undergoing AHT only includes those who are already in follow-up care. Patients undergoing endocrine induction therapy were not included in this group.

The empirical distribution of the age group disease stage, and CHT- and AHT-status does deviate significantly from a normal distribution. Relevant for the subsequent analyses, however, is, primarily, the assumption of a normal distribution of BMI and physical activity at baseline.

The empirical distribution of baseline BMI (chi-square(5, *n* = 776) = 195.4, *p* = 1.00), steps per day at baseline (chi-square(5, *n* = 750) = 1.989, *p* = 1.00), and physical activity at baseline (chi-square(5, *n* = 706) = 1.053, *p* = 0.987) does not deviate significantly from a normal distribution.

### 3.2. Body Mass Index 

Table 2 displays the BMI data for the entire cohort as well as in the examined subgroups of patients undergoing AHT and CHT. 

In the comparison of means, a reduction of one BMI point over 6 months was observed in the subgroup of CHT patients and in the subgroup of AHT patients with a BMI over 30, which corresponds to a significant reduction with *p* < 0.001. During AHT, after the first 12 weeks, 41.4% of patients experienced weight gain, 33.4% were able to maintain their weight, and 24.2% reduced their weight.

A repeated measures ANCOVA with a Greenhouse–Geisser correction determined that mean performance levels showed a statistically significant difference between measurements [*F*(1.47, 295.5) = 7.46, *p* = 0.02] for BMI overall and [*F*(1.47, 295.5) = 3.44, *p* = 0.048] for BMI and CHT status. 

AHT, disease stage (EBC/MBC), and therapy status (in therapy/follow-up) did not yield significant results. Bonferroni-adjusted post-hoc analysis for the dataset from baseline to T1 (12 weeks) revealed a significant BMI reduction in the subgroup that did not receive CHT (*p* = 0.03) and the subgroup under acute CHT (*p* = 0.01) with *M*_Diff_ = 1.66, 95%-CI [0.19, 3.13]. For the post-hoc test, the number of patients at time point T2 (24 weeks) was too small. The CHT subgroups with CHT completed, unknown, and planned CHT did not show significant changes.

### 3.3. Physical Activity (Steps per Day and Activity Minutes)

The following table (Table 3) presents the results of physical activity showing the average number of steps per day over the measurement period and the monthly average of activity minutes per week. The *p*-values, *F*, and standard deviations are reported. Additionally, the two relevant subgroups of patients under AHT and CHT were analyzed.

The number of steps per day is determined through the app, while the patient self-reports the activity minutes. At the 6-month mark (T2), there is a lower number of datasets because not all patients in the examined cohort had been using the app for more than 3 months (T1).

A repeated measures ANCOVA with a Greenhouse–Geisser correction determined that mean performance levels showed a statistically significant difference between measurements, [*F*(1.91, 713.5) = 7.46, *p* = 0.01] for mean steps/day overall, and [*F*(1, 231) = 26.86, *p* < 0.01] for weekly activity minutes. CHT, AHT, disease stage (EBC/MBC), and therapy status (in therapy/follow-up) did not yield significant results in a post-hoc test.

## 4. Discussion

Although randomized controlled trials are commonly regarded as the “gold standard” in medical research, they frequently do not accurately mirror the behaviors of actual patients and the resulting outcomes in real-world clinical practice. Due to the significant impact of patient attitudes and behaviors on breast cancer self-management, alternative research approaches are essential to conclusively evaluate behavior-based interventions, such as the utilization of mHealth apps in real-world settings.

In the context of this retrospective observational study, real-world data on BMI and physical activity were collected from PINK! Coach users. At the outset, we investigated a cohort of patients excluded from the analyses. There were no notable differences noted between the two sets concerning clinical parameters, including age, disease stage, tumor stage, therapy status, and CHT- and AHT-status.

The empirical distribution of age group, disease stage, and CHT- and AHT-status does deviate significantly from a normal distribution. Relevant for the subsequent analyses, however, is, primarily, the assumption of a normal distribution of BMI and physical activity at baseline.

Across the entire cohort, a consistent BMI was observed over 24 weeks. Given the existing literature on breast cancer patients undergoing CHT or AHT, these results are clinically relevant [47,48,49,50].

Within BMI subgroups (BMI < 25, BMI 25–30, and BMI > 30), BMI maintenance was also evident. The subgroup of patients with a BMI over 30 undergoing AHT exhibited a one-point improvement in BMI over 24 weeks, indicating a significant reduction. The overall group of patients undergoing CHT also demonstrated a one-point reduction in BMI, representing a significant decrease [28,51,52,53]. In the AHT subgroup, there were *n* = 113 patients at the baseline and T1 time points. In the subgroup of patients undergoing CHT, *n* = 125. The number of patients at the T2 time point after 6 months decreased significantly in both groups. The PINK! Coach App is approved as a digital health application and patients only receive a prescription for 3 months at a time. Since only a smaller proportion of patients obtain a second prescription, resulting in data points after 6 months, the number of patients decreases. This should be considered as a limitation in interpreting the results. Nevertheless, from a clinical perspective, the goal is for patients receiving cortisone or undergoing AHT to maintain their weight.

A repeated measures ANCOVA with a Greenhouse–Geisser correction determined a significant association between CHT status and BMI change. The subsequent post-hoc test revealed a significant reduction in BMI in the CHT subgroups ‘under acute therapy’ and ‘no therapy’. The three remaining groups, ‘therapy planned’, ‘therapy planned’, and ‘not yet known’, showed no changes in the small numbers of patients.

Current research studies indicate that patients undergoing acute CHT or AHT typically experience an average weight gain of over 5% within 6 months [47,48,49,50]. In our cohort, approximately 58% of patients undergoing AHT and 54% of patient undergoing CHT were able to either maintain or even reduce their weight during the first 12 weeks of using the app. From a clinical perspective, the results of the PINK! Coach App are highly relevant in the subgroups of CHT and AHT as weight gain can be expected due to hormonal therapy or the administration of corticosteroids during CHT.

It is conceivable that these results can be attributed to the educational functions as well as the everyday lifestyle coaching provided by the app.

The study data also indicates a significant increase in physical activity (steps per day and activity minutes per week) in the overall cohort as well as in the AHT subgroup. The CHT patient subgroup shows a trend but is not statistically significant (steps/day: *p* = 0.06, and weekly activity minutes: *p* = 0.07). These results align with the BMI data findings. Physical activity, alongside nutrition, plays a crucial role in terms of weight management. Since the CHT group slightly increased their physical activity but still showed a reduction in BMI, it appears that the patients’ nutrition plays an important role. For future research projects, it would be interesting to examine detailed nutritional records of the patients. The results regarding physical activity also indicate that none of the included confounders had an impact on the data. Consequently, it can be presumed that all patients benefited.

However, this study has several limitations that should be considered. Firstly, this study did not show data from a control group as it is a data set exported from the PINK! Coach App in a real-world setting. This can limit the meaningfulness of the results.

Secondly, self-documented data from the patients may have limited reliability, and there are additional relevant confounders that were not included in this study as they were not collected, including menopausal status and socio-economic parameters. For future analyses, collecting patient-reported outcomes, such as quality of life and fatigue, would be clinically relevant. Since the step count of the patients was determined through the app and may not have always been carried with them, this could result in a lower step count than actually taken. Nevertheless, patients were encouraged to carry their smartphones when moving.

Furthermore, the observation period is very short. Clinically relevant are the 2–5 years following the completion of acute therapy, during which most patients undergo AHT. During this period, nutrition and exercise play an essential role [15,54,55,56,57,58].

## 5. Conclusions

The data presented in this paper are the first real-world data from the only officially approved Digital Health Application (DiGA) in Germany for breast cancer patients. The data provide initial insights into the real-world care of patients undergoing acute therapy and survivorship. It is conceivable that these results can be attributed to the educational functions as well as the everyday lifestyle coaching provided by the PINK! Coach App.

Further analyses are necessary to better understand the impact of coaching through the app PINK! Coach. Long-term data spanning 12 months and more could provide an even deeper insight into the app’s long-term effectiveness. 

## Figures and Tables

**Figure 1 cancers-16-01020-f001:**
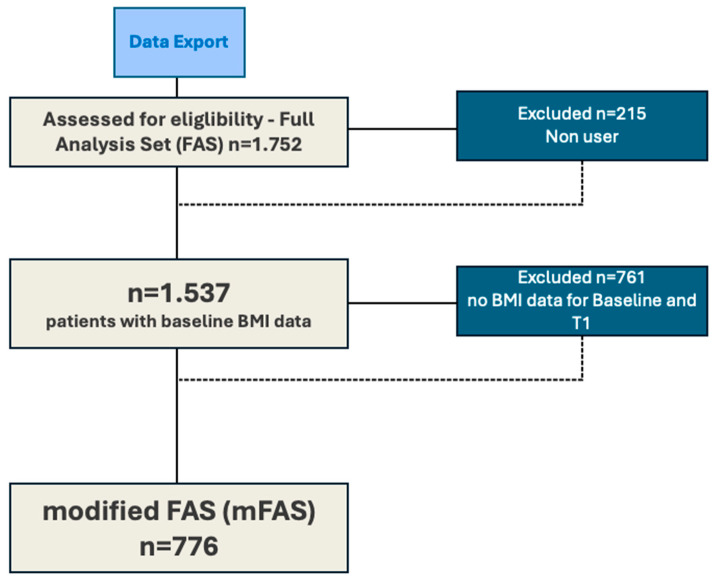
Patient Flow-Chart with number of patients assessed for eligibility and mFAS.

**Table 1 cancers-16-01020-t001:** Characteristics of the eligible patients mFAS.

Parameter		mFAS Number of Patients *n* (%)	Pearson Chi-Square χ^2^ (*p*)
overall		776 (100.0)	633.43 (*p* < 0.01)
Age groups	18–29	3 (0.4)	
	30–39	43 (5.5)	
	40–49	165 (21.2)	
	50–64	458 (59.0)	
	65+	107 (13.8)	
mean age	52.6	776 (100.0)	
Disease stage	EBC *	694 (89.4)	482.66 (*p* < 0.01)
	MBC **	82 (10.6)	
Therapy status	In acute therapy	367 (47.3)	2.27 (*p* = 0.132)
	Survivor/Follow-up	409 (52.7)	
CHT-status	No CHT ***	316 (40.7)	
	Under acute CHT	125 (16.1)	
	CHT completed	324 (41.8)	
	Unknown	6 (0.8)	
	Planned CHT	5 (0.6)	
AHT-status	No AHT ****	179 (23.1)	
	Under acute AHT	443 (57.1)	
	AHT completed	27 (3.5)	
	Unknown	31 (12.4)	
	Planned AHT	96 (12.4)	

* Early breast cancer (EBC); ** metastatic breast cancer (MBC); *** Chemotherapy; **** anti-hormone therapy.

**Table 2 cancers-16-01020-t002:** Trends in BMI after implementation of PINK! Coach in mFAS.

	mFAS		Subgroup AHT ***		Subgroup CHT **	
Overall	Mean (±SD)	*n*	Mean (±SD)	*n*	**Mean (±SD)**	*n*
Baseline	25.6 (±5.2)	776	26.1 (±5.6)	447	**24.8 (±1.9)**	**125**
12 weeks	25.3 (±4.9)	776	25.8 (±5.3)	447	**24.3 (±2.2)**	**125**
24 weeks	25.1 (±4.5)	215	25.8 (±4.9)	126	**23.8 (±2.4) ***	**68**
BMI < 25						
Baseline	22.0 (±1.8)	401	22.0 (±1.8)	220	21.8 (±2.0)	73
12 weeks	21.9 (±1.9)	401	21.9 (±1.9)	220	21.7 (±2.2)	73
24 weeks	22.0 (±2.0)	113	22.0 (±1.7)	40	21.9 (±2.4)	21
BMI 25–30						
Baseline	27.1(±1.4)	223	27.1(±1.4)	114	26.9 (±1.4)	32
12 weeks	26.7 (±1.6)	223	26.8 (±1.5)	114	26.4 (±1.8)	32
24 weeks	26.5 (±1.9)	28	26.5 (±1.8)	32		
BMI > 30						
Baseline	33.9 (±4.6)	152	**34.3 (±5.1)**	**113**	32.7 (±2.5)	20
12 weeks	33.4 (±3.7)	152	**33.5 (±3.8)**	**113**	32.2 (±2.2)	20
24 weeks	33.1(±3.1)	74	**33.3 (±3.4) ***	**54**		

***** bold data points indicate a significant reduction with *p* < 0.01; ** Chemotherapy; *** anti-hormone therapy.

**Table 3 cancers-16-01020-t003:** Physical activity recorded with PINK! Coach measured in mean steps per day and physical activity minutes per week.

Mean	Steps (*n*)	*p*-Value (F) (Baseline-T2)	Standard Deviation SD	Subgroup AHT *** (*n*)	*p*-Value (Baseline-T2)	Standard Deviation SD	Subgroup CHT ** (*n*)	Standard Deviation SD	*p*-Value (Baseline-T2)
Daily steps first 4 weeks	4833 (750)	0.01 (7.46)	±2913	5061 (426)	<0.001	±2962	4261 (119)	±2792	0.06
Daily steps 12 weeks	5388 (731)		±3180	5633 (419)		±3293	4474 (113)	±2873	
Daily steps 24 weeks	5384 (380)		±3026	5579 (227)		±3185	4506 (33)	±2856	N/A *
Weekly activity minutes 4 weeks	61.9 (706)	<0.01 (26.86)	±27.3	63.8 (404)	<0.001	±37.7	53.9 (110)	±28.2	0.07
Weekly activity minutes sports 12 weeks	64.3 (575)		±25.4	68.9 (341)		±45.5	55.0 (83)	±27.6	
Weekly activity minutes 24 weeks	71.3 (268)		±24.9	75.4 (166)		±63.7	57.1 (24)	±30.0	N/A *

* N/A number of patients in this subgroup was too small; ** Chemotherapy; *** anti-hormone therapy.

## Data Availability

The data is not provided due to commercial and copyright reasons.

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
