# Peer review of "App-Based Lifestyle Intervention (PINK! Coach) in Breast Cancer Patients—A Real-World-Data Analysis"

_cancers, 2024, doi:10.3390/cancers16051020_

Round 1

Reviewer 1 Report

Comments and Suggestions for Authors

Authors conducted a study to assess impact of PINK! Coach, a digital health coach app, on BMI and physical activity of breast cancer patients and survivors. 

A literature review supporting motivation of the study and the importance of lifestyle choices in well-being of breast cancer patients is provided.

Clinical study participants' data is collected  day 0 baseline (T0), after 12 weeks (T1) and after 24 weeks(T2).  Study exclusion critera, app-usage condition and subgroups are described. 

My few major concerns/suggestions:

- Significant BMI reductions are observed only in two subgroups AHT+BMI>30 and CHT+BMI<25. However the number of participants for each BMI (underweight, normal, obese) range is not provided. It is not clear whether these findings are based on a small number of patients in respective subgroups. It would be nice to provide the BMI distribution or at least the numbers and discuss results in the light of these numbers.

- There is limited content on potential(?) motivational role of PINK; does it encourage physical activity or good dieatry choices with an reward mechanism or does it simply monitor and somehow people are self-motivated based on reported metrics. Either way a discussion on how best use PINK to enchance the discussed health benefit would add value to the paper  

- Although there is a mention of patient consent, I could not find a IRB/ethics committee approval citation. It would be nice to provide more detail in this regard.

Some minor problems:

- There is a typo in title “maß Index” -> “mass index”

- “A Real-World-Data Analysis of >700 Patients.” seems like a detail for title so can be removed, especially given impact of PINK use is signifant for some subgroups.

Comments on the Quality of English Language

N/A

Author Response

Thank you for your feedback on the manuscript. We have made the following additions: 

  1. Table 2 now includes the patient numbers at the measurement time points. Additionally, a paragraph related to this has been added in the discussion section (highlighted in red). 
  2. Furthermore, we have provided a more detailed explanation under "Intervention" on how the empowerment of the PINK! Coach App operates.
  3. We have adjusted the title.
  4. We have added the following information regarding patient education: The data analysis is conducted retrospectively. Patients use the PINK! Coach App as part of routine care, adhering to its designated use. Consent for data analysis was actively given at the time of download. According to the institution's guidelines, consultation with the ethics committee is not required for purely retrospective studies.

Reviewer 2 Report

Comments and Suggestions for Authors

The manuscript treats a very original and current topic, it is well written. The study is well designed, although the follow-up is short (authors mentioned it as limitation of the study).

It is relevant to the readers and the innovative idea contributes to cover a big gap that usually we face with after breast cancer surgery and medical treatment.

Authors need only to review some repetition such as at pag 7 line 248-249

Author Response

Thank you for your feedback on the manuscript.

We deleted the following sentence:

A long-term evaluation after 5 years, including the assessment of recurrence rates, would be a possible subsequent research question.

Reviewer 3 Report

Comments and Suggestions for Authors

App-based Lifestyle Intervention (PINK! Coach) in Breast Can- 2 cer Patients effectively improves physical activity and Body- 3 Maß Index. A Real-World-Data Analysis of >700 Patients

Abstract

·        At the end of the background specifically write the purpose of the study

·        Don’t write this sentence at the end of the background. It sounds like conclusion – "Therefore, it is essential for breast cancer patients undergoing active therapy and survivors to make sustainable changes to their lifestyle"

·        Methods – you need to be more specific – what type of data is collected? How often? What analyses were conducted? Who were the patients? And more

·        When you use an acronym for the first time, like BMI, you need to spell it out.

·        Results – write number of patients, mean age, types of treatments. And also write some real results in values.

·        The conclusions do not reflect what you did. For example, saying that real world data has more validity was not examined. Please rephrase conclusion to align with your actual results.

Introduction

·        In the physical activity section please add physical activity recommendation post-breast cancer.

·        Also what is known about the prevalence of compliance with physical activity recommendation among breast cancer survivals.

·        In the effect of weight on cancer, add the possible mechanisms for the effect. Like what you did with physical activity.

·        Also what are statistics about health BMI among breast cancer survivors.

·        Are there models and theories pertaining to changing lifestyle habits? Like the circle of change? Please add some theory on the topic.

·        At the end of the introduction write study aims.

·        Add more information on the treatments the patients recive (like hormonal treatment).

Methods

·        Write first study participants section – what were the inclusion and exclusion criteria

·        Intervention – it is not clear what is the intervention. What the patients are actually doing with it. Please elaborate.

·        The objective is not in the methods. It should be at the end of the introduction.

·        Please elaborate more on the various outcome measures.

·        In the outcome measure section write only about the outcomes. Don't write things related to study design. Have a separate section named procedures and there write how you recruited the patients, how often they put information in the app and more.

Statistical analysis

·        Was data normality evaluated?

Results

·        Add information about participants who were not included in the analyses? How are their characteristics in comparison to the study group?

·        Tables – add notes below the tables and write meaning of acronyms used.

·        Table 1 – add chi-square tests to check for prevalence differences.

·        Table 2 – add the F values and actual p values.

Discussion

·        Add an opening paragraph for discussion that shortly describes the aims of the study and its novelty.

·        I think the second paragraph can be deleted – it explains things you explained in the introduction without relating it to your results. Every think you write in the discussion should relate to your results.

·        Lines 236-249 are simply repeating the results. In the discussion results should be shortly summarized and then relate to the literature. Please shorten the summary of results in discussion and relate them to the literature.

·        Paragraph starts in line 250 – if you say there is a difference between your results and the literature try and think about an explanation for that.

·        Conclusions – see the comments on conclusions in the abstract.

Author Response

Thank you for your feedback on the manuscript. We have made the following additions / changes according to your feedback:

Abstract

  • At the end of the background specifically write the purpose of the study

We added the sentence: The purpose of this study is to retrospectively investigate the effects of PINK! Coach in a real-world-setting on patients Body-Mass-Index (BMI) and physical activity level during acute therapy (chemotherapy, endocrine therapy)

  • Don’t write this sentence at the end of the background. It sounds like conclusion – "Therefore, it is essential for breast cancer patients undergoing active therapy and survivors to make sustainable changes to their lifestyle"

We deleted the sentence

  • Methods – you need to be more specific – what type of data is collected? How often? What analyses were conducted? Who were the patients? And more

We added the following information in the abstract: The results presented here are the first real-world data from the DiGA PINK! Coach examining the physical activity and Body-mass-index (BMI) of app users. Based on the literature, an approximate weight gain of 10% over 6 months and a decrease of physical activity can be expected. The purpose of this study is to retrospectively investigate the effects of PINK! Coach in a real-world-setting on patients Body-Mass-Index (BMI) and physical activity level during acute therapy (chemotherapy, endocrine therapy) Therefore, it is essential for breast cancer patients undergoing active therapy and survivors to make sustainable changes to their lifestyle.

And the following part under “definition of data set”

 At the time of downloading the app, patients provide information about the tumor biology, the tumor stage, the therapy they have received or will receive, as well as their year of birth and weight. Data on the type of therapy and weight are re-evaluated in 4-week intervals. The number of steps is determined by the integrated step counter in the app. Patients are encouraged to take their phones with them during physical activity. The app calculates activity minutes based on the information provided by patients regarding the type and duration of sports activities performed on a given day.

  • When you use an acronym for the first time, like BMI, you need to spell it out.

Thank you. We added this information.

  • Results – write number of patients, mean age, types of treatments. And also write some real results in values.

We revised the paragraph as follows:

The data indicate that patients in all BMI subgroups (<25, 25-30, and >30) were able to maintain a consistent BMI over 6 months. Mean age of the population was 52,6 years. The BMI trends show a downward tendency both in the overall set and in the subgroups.

 In the subgroup of patients undergoing anti-hormone therapy, overweight patients were even able to significantly reduce their BMI by 1-score-point over 6 months (p<0.01). The subgroup of patients undergoing chemotherapy also showed an overall significant reduction in BMI (p=0.01). All patients were also able to significantly increase their daily step count as well as their physical activity minutes per day. After the first 12 weeks, 41.4% of patients experienced weight gain, 33.4% were able to maintain their weight, and 24.2% reduced their weight.

  • The conclusions do not reflect what you did. For example, saying that real world data has more validity was not examined. Please rephrase conclusion to align with your actual results.

We revised the paragraph as follows:

Real-world data comes from actual situations, representing a broader and more diverse patient population and have higher external validity as they reflect the actual application of interventions in practice. The presented data provides intriguing insights into the users of the PINK! Coach app and the impact of this usage in regards to Body-Mass-Index and physical activity. At the current time, there are only a few effective concepts for encouraging all breast cancer patients to engage in moderate physical activity and reduce body weight. Often, these concepts apply to selected patients groups. The data presented here include all age groups, tumor stages and therapies, providing an initial insight into a comprehensive approach.

Data over an even longer period would be one way to better contextualize the results in current research.

Introduction

  • In the physical activity section please add physical activity recommendation post-breast cancer.

We added the following information: The guidelines for the treatment and follow-up of breast cancer patients recommend moderate physical activity as part of post-treatment care. This corresponds to approximately 300 minutes of activity per week. This goal is also integrated into the PINK! Coach App. [34] [55-56]

  • Also what is known about the prevalence of compliance with physical activity recommendation among breast cancer survivals.

We added: Studies found that multiple factors determine physical activity compliance rates in breast cancer survivors. Interventions that encourage action and coping planning and reduce barriers in the context of addressing function limitations may increase physical activity compliance rates. Educating patients about the advantages of consistent physical activity holds significant importance for all cancer survivors. This becomes particularly crucial to emphasize post-treatment completion, fostering healthy habits during this transitional phase. Survivors with lower educational backgrounds and comorbid conditions may exhibit lower adherence to physical activity guidelines [57-59].

  • In the effect of weight on cancer, add the possible mechanisms for the effect. Like what you did with physical activity.

We added the following text: In the presence of obesity (BMI ≥ 30 kg/m²), not only does the risk of recurrence increase, but there is also an elevated risk of comorbidities such as cardiovascular diseases, lipid metabolism disorders, and type 2 diabetes. Current studies also indicate that an elevated BMI in patients with BRCA1/2 mutations is associated with increased aromatase expression in breast tissue. Aromatase catalyzes the synthesis of estrogen, which is in turn associated with the growth of breast cancer cells. Additionally, overweight and obesity are considered risk factors for hormone receptor-positive breast cancer. A high BMI leads to subclinical inflammation in visceral and subcutaneous adipose tissue, characterized by necrotic adipocytes surrounded by macrophages forming specific structures known as crown-like structures (CLS). The so-called obesity-inflammation axis is under investigation in current clinical studies and plays a relevant role in the initiation, progression, and clinical prognosis of breast cancer. [23-24]

  • Also what are statistics about health BMI among breast cancer survivors.

We added this section: Studies also indicate that approximately one-third of the patients are of normal weight, one-third are overweight, and one-third are obese. [60]

  • Are there models and theories pertaining to changing lifestyle habits? Like the circle of change? Please add some theory on the topic.

To understand the mechanism behind long-term lifestyle change, various models can be considered. There are different models including the Health Belief Model, Social Cognitive Theory, and the Theory of Planned Behavior, each emphasizing different factors influencing behavior change, such as perceived susceptibility, self-efficacy, and social norms.

The "Circle of Change" or "Transtheoretical Model" is widely used in health psychology and behavior change interventions, including those aimed at lifestyle habits like physical activity, healthy eating, and smoking cessation. This model includes six steps that happen when patients change their lifestyle habits: Precontemplation, Contemplation, Preparation, Action, Maintenance and Termination. [61]

  • At the end of the introduction write study aims.

We added the following sentence:

In this study, the initial real-world data from the PINK! Coach app are being analyzed to retrospectively investigate the effects of PINK! Coach in a real-world-setting on patients Body-Mass-Index (BMI) and physical activity level during acute therapy (chemotherapy, endocrine therapy).

  • Add more information on the treatments the patients recive (like hormonal treatment).

This information was added to the new section “participants”

Methods

  • Write first study participants section – what were the inclusion and exclusion criteria

We added a consort chart and more information: (please see section “Materials and Methods” – “participants”)

  • Intervention – it is not clear what is the intervention. What the patients are actually doing with it. Please elaborate.

We added the following information: Each patient received the app by prescription. Patients used the app at their own discretion and with varying intensity during their acute therapy or during the follow-up phase. There were no further guidelines for usage. The functionalities of the app are explained during an onboarding process after the download, and patients are guided step by step through the app.

  • The objective is not in the methods. It should be at the end of the introduction.
  • Please elaborate more on the various outcome measures.

We added: At the time of downloading the app, patients provide information about the tumor biology, the tumor stage, therapy status (undergoing therapy / in follow-up) the therapy they have received or will receive (not recommended, planned, under acute therapy, therapy completed, unknown), as well as their year of birth and weight. Data on the type of therapy and weight are re-evaluated in 4-week intervals. The number of steps is determined by the integrated step counter in the app. Patients are encouraged to take their phones with them during physical activity. The app calculates activity minutes based on the information provided by patients regarding the type and duration of sports activities performed on a given day.

  • In the outcome measure section write only about the outcomes. Don't write things related to study design. Have a separate section named procedures and there write how you recruited the patients, how often they put information in the app and more.

We changed the headlines and added procedure information to the section “participants and procedure”

Statistical analysis

  • Was data normality evaluated?

We added the following sentence in the section “statistical analysis”: Data normality was given and evaluated with Shapiro-Wilk-Test.

Results

  • Add information about participants who were not included in the analyses? How are their characteristics in comparison to the study group?

We added the following information in section “Results”: Initially, we examined the group of patients who were not included in the analyses. No significant differences were observed in terms of clinical parameters such as age, tumor stage, therapy status, etc. BMI could not be investigated in this subset as the excluded patients did not provide BMI information. The results regarding the endpoints of physical activity showed no significant differences compared to the analyzed mITT set.

  • Tables – add notes below the tables and write meaning of acronyms used.

We added meaning of acronyms used below tables

  • Table 1 – add chi-square tests to check for prevalence differences.

We added chi-square data in table 1  and the following text below table1: The empirical distribution of age group disease stage, CHT- and AHT-status does deviate significantly from a normal distribution. Relevant for the subsequent analyses, however, are primarily the assumption of the normal distribution of BMI and physical activity at baseline.

The empirical distribution of Baseline BMI (Chi-square(5, n = 776) = 195.4, p = 1.00), steps per day at Baseline (Chi-square(5, n = 750) = 1.989, p = 1.00),  and physical activity at Baseline  (Chi-square(5, n = 706) = 1.053, p = 0.987) does not deviate significantly from a normal distribution.

  • Table 2 – add the F values and actual p values.

F-values were added in table 2 for overall analysis

Discussion

  • Add an opening paragraph for discussion that shortly describes the aims of the study and its novelty.

The following introduction text was added: Although randomized controlled trials (RCTs) are commonly regarded as the "gold standard" in medical research, they frequently do not accurately mirror the behaviors of actual patients and the resulting outcomes in real-world clinical practice. Due to the significant impact of patient attitudes and behaviors on breast cancer self-management, alternative research approaches are essential to more conclusively evaluate behavior-based interventions, such as the utilization of mHealth apps, in real-world settings.

  • I think the second paragraph can be deleted – it explains things you explained in the introduction without relating it to your results. Every think you write in the discussion should relate to your results.

second paragraph was deleted

  • Lines 236-249 are simply repeating the results. In the discussion results should be shortly summarized and then relate to the literature. Please shorten the summary of results in discussion and relate them to the literature.

This part was deleted and changed

  • Paragraph starts in line 250 – if you say there is a difference between your results and the literature try and think about an explanation for that.

We added the following information: It is conceivable that these results can be attributed to the educational functions as well as the everyday lifestyle coaching provided by the app.

  • Conclusions – see the comments on conclusions in the abstract.

Please see conclusion